# Polymorphisms and mRNA Expression Levels of *IGF-1*, *FGF5,* and *KAP 1.4* in Tibetan Cashmere Goats

**DOI:** 10.3390/genes14030711

**Published:** 2023-03-14

**Authors:** Tianzeng Song, Yao Tan, Renqing Cuomu, Yacheng Liu, Gui Ba, Langda Suo, Yujiang Wu, Xiaohan Cao, Xianyin Zeng

**Affiliations:** 1Institute of Animal Science, Tibet Academy of Agricultural and Animal Husbandry Science, Lhasa 850009, China; 2Isotope Research Laboratory, Sichuan Agricultural University, Ya’an 625014, China

**Keywords:** mRNA expression, SNP loci, Tibetan cashmere goat

## Abstract

The Tibetan cashmere goat is a precious breed in China and its cashmere is widely used in clothing and textiles. The genes *IGF-1*, *FGF5*, and *KAP 1.4* have been shown to be crucial regulators of cashmere growth. In this study, we examined mRNA expression levels of these three genes and detected *IGF-1*, *FGF5*, and *KAP 1.4* SNP loci in the Tibetan cashmere goat. After amplification and sequence alignment of the genes *IGF-1*, *FGF5*, and *KAP 1.4* among 206 Tibetan cashmere goats, two new SNP loci were detected in gene *KAP 1.4*, while no SNP loci were found in amplified fragments of genes *IGF-1* and *FGF5*. The expression levels of gene *IGF-1* in Baingoin and Nyima counties were significantly higher than in other counties (*p* < 0.05). Moreover, the expression level of gene *FGF5* in Gêrzê was significantly higher than in Rutog. The expression levels of mRNA in *KAP 1.4* showed significant variation among seven counties. There were no significant differences in mRNA expression levels of *IGF-1*, *FGF5*, and *KAP 1.4* in Tibetan cashmere goats when analysed by sex. The gene *IGF-1* was slightly up-regulated in one to five-year-old cashmere goats, except in those that were 4 years old. The mRNA expression levels of *FGF5* in one and two-year-old cashmere goats was lower compared with those in three to five-year-old cashmere goats. *KAP 1.4* was up-regulated across one to five-year-old cashmere goats. In this study, SNP detection and mRNA expression analysis of *IGF-1*, *FGF5*, and *KAP 1.4* genes was able to add data to genetic evolutionary analysis. Further studies should be carried out in SNPs to detect other fragments in genes *IGF-1* and *FGF5,* as well as signal pathways and gene functions in protein levels of genes *IGF-1*, *FGF5*, and *KAP 1.4* in the Tibetan cashmere goat.

## 1. Introduction

Goats (*Capra hircus*) have long been raised to meet a variety of essential human needs. The demand for cashmere is growing in accordance with the development of humanity. The Tibetan cashmere goat is a well-known domestic breed in the northwestern region of Tibet, China, and is renowned for its cashmere [1,2]. The Tibetan cashmere goat is well adapted to the temperature and harsh surroundings of the Tibetan Plateau. The wool from this breed has a distinct, soft texture and a high commercial worth [3]. Genetic variation in genes such as *KRT* and *KAP* in different breeds of cashmere goats contributes to phenotypic differences. Different hair fiber-associated genes are involved in determining skin architecture and hair development. The diversity in these genes may regulate the structure and traits of wool. The growth of cashmere is also regulated by hair follicles, which include the primary hair follicle and the secondary hair follicle. Wool fiber is derived from primary hair follicles, whereas cashmere is derived from secondary hair follicles [4,5].

Previous studies have found that the genes *IGF-1*, *FGF5*, and *KAP 1.4* are all linked to cashmere development [6,7,8]. These three genes have been investigated in other Chinese domestic goat breeds, including the Liaoning [9,10] and Inner Mongolian cashmere goats [11,12]. However, we are aware of only a few research studies involving Tibetan cashmere goats. *IGF-1* is structurally homologous to proinsulin and acts as a polypeptide hormone. *IGF-1* is essential for mammalian growth, development, and homeostasis. It is an essential growth hormone regulator [13]. Previous studies have shown that *IGF-1* contains at least six exons in many mammalian species and complicated alternative splicing [10,14,15,16] and multiple nuclear polymorphic sites exist in this gene. Polymorphisms in the *IGF-1* gene have been linked to developmental characteristics and yearling hair fiber weight in Iranian domestic cashmere goats [6] and Nanjiang Huang goats [17].

The *FGF5* gene, which was initially discovered as an oncogene, has been suggested to be involved in human and mouse hair follicle cycles [18,19]. The *FGF5* gene can generate two types of proteins: a full-length protein (*FGF5*) and a short protein (FGFs) [18]. A previous study found that the human hair development cycle enters the catagen phase prematurely due to an elevated expression level of *FGF5* [19]. This indicates that *FGF5* may inhibit hair follicle development [20]. Indeed, an autosomal recessive mutation in the *FGF5* gene in the Angora mouse resulted in an increase in hair development [21]. Furthermore, a missense mutation in the *FGF5* gene in dogs resulted in longer hair [22].

Keratin-associated protein (KAP) 1.4 is a primary component of hair fiber. KAP is a large protein family that is divided into three types based on amino acid content: high glycine–tyrosine KAPs, high-sulfur KAPs, and ultra-high-sulfur KAPs [8,12]. The protein levels of KAP in various species is different. Previous research found that the *KAP* gene family has many polymorphisms, and there are many SNP sites in goat and sheep breeds [8,12,23,24].

Sheep *KAP 1.4* has a 30-nucleotide deletion when compared to *Ovis aries* and *Capra hircus* [8]. *KAP 1.4* is a high-sulfur KAP, and it has been demonstrated that *KAP 1.4* has a relationship with the quantity as well as quality of animal hair fibers. Variations in the KAP family play essential roles in the hair development cycle. The aim of this study was to determine whether *IGF-1*, *FGF5*, and *KAP 1.4* are related to cashmere development in Tibetan cashmere goats through mRNA expression and polymorphisms.

## 2. Materials and Methods

### 2.1. Experimental Animals

A total of 240 cashmere goats of various sexes and ages were chosen at random from seven counties in the Ngari and Nagqu provinces of Tibet. A map showing counties and altitudes is presented in Figure 1. A total of 480 skin tissue samples (1 cm^3^) were collected from the ears of living goats in October (two ear tissue samples were obtained from each goat) for DNA and RNA extraction. All tissue samples were frozen in liquid nitrogen before being stored at −80 °C until use, and tests were carried out with the permission of the Sichuan Agricultural University Animal Experimental Committee (SAU2015064, 12 December 2015).

### 2.2. DNA and RNA Extraction and cDNA Synthesis

Genomic DNA was extracted using the TIANamp Genomic DNA kit (TIANGEN, Bio, Co., Ltd. Beijing, China) according to the manufacturer’s instructions. Total RNA was extracted using the phenol chloroform extraction technique using RNAiso Plus (TaKaRa Bio, Co., Ltd., Dalian, China). To analyze the mRNA expression level, cDNA was synthesized using the PrimeScript™ RT reagent Kit with gDNA Eraser (TaKaRa Bio, Co., Ltd., Dalian, China) according the manufacturer’s instructions. The instructions for cDNA synthesis were as follows: 37 °C for 15 min and 85 °C for 5 s in a volume of 20 μL with a total of 1 μg RNA isolated from ear skin tissue, 5 × gDNA Eraser buffer, gDNA Eraser, PrimeScript RT Enzyme Mix I, RT Primer Mix, 5 × PrimeScript Buffer 2, RNase-free dH_2_O. The primers used for cDNA synthesis were Oligo dT Primer with Random 6-mers.

### 2.3. PCR and DNA Sequencing

Three-pair primers were designed and synthesized to amplify the desired fragments of *IGF-1*, *FGF5*, and *KAP 1.4* (Table 1). The primers were designed using partial exon segments from previously published data [17,18,25]. PCR amplification was performed in a volume of 25 μL in a T100 Thermal Cycler (Bio-Rad Laboratories, Inc., Shanghai, China), using the Premix Taq™ kit and genomic DNA as a template, with the following reaction conditions: 3 min denaturation at 95 °C, followed by 35 cycles (30 s at 94 °C, 40 s at the corresponding annealing temperature, and 24 s at 72 °C for extension), ending with an extension of 5 min at 72 °C. After electrophoresis on 2% agarose gel, PCR products were visualized using the UV trans-illuminator (Bio-Rad Laboratories, Inc.) and were purified for sequencing (Sangon Biotech, Co., Ltd., Shanghai, China).

### 2.4. Genotyping and Genetic Analysis

SNPs were detected using the sequencing technique, and the sequence patterns of 206 DNA samples were examined (due to long-distance transport or accidents, 34 ear tissue samples were damaged and not able to be used in this experiment). Based on the sequence data, allele frequencies, genotype numbers, heterozygosity, and homozygosity were determined. MEGA 6.06 and DNAMAN8 software (LynnonBiosoft, San Ramon, CA 94583, USA) were used for sequence alignment, similarity analysis, statistical genetics, and phylogenetic tree analysis. A phylogenetic tree was constructed using the sequence of the genes *IGF-1*, *FGF5*, and *KAP 1.4* in Tibetan cashmere goats and other animals.

### 2.5. Quantitative PCR for mRNA Expression

The mRNA expression levels of *IGF-1*, *FGF5*, and *KAP 1.4* in Tibetan cashmere goats were determined using quantitative PCR (qPCR) with cDNA as a template, which was reverse-transcribed from total RNA. The qPCR was carried out in triplicate using the CFX96 Real-Time PCR detection system (Bio Rad, Hercules, CA, USA) with the use of the SYBR^®^Premix Ex Taq™ II kit (TaKaRa Bio, Co., Ltd., Dalian, China). The qPCR procedure was as follows: denaturation at 95 °C for 30 s, followed by 35 cycles of reaction (95 °C for 5 s, annealing at the corresponding temperature for 30 s). In this study, a fluorescent signal was detected at the annealing temperature, and melting curve analysis was performed at the end of the qPCR reaction to assess the specificity of the PCR product. The relative genes’ expression levels were normalized to *β-actin*. The detailed information of primers used is shown in Table 1.

### 2.6. Statistical Analysis

The DNAMAN 8 and MEGA 6.03 statistical systems were used to evaluate target gene sequences and SNP detection. Sequence alignment was calculated using the discovered SNP loci, gene frequencies, allele frequencies, heterozygosis, homozygosis, and phylogenetic tree [9,18,26]. Relative mRNA expression levels of *IGF-1*, *FGF5*, and *KAP 1.4* were analyzed using the GraphPad prism 5 program (GraphPad prism software, Inc., Boston, MA, USA). Differences in the expression levels of target genes in various groups were analyzed using the one-way ANOVA. Results were expressed as *X* + SD and triplicate [7,10].

## 3. Results

### 3.1. Cloning of the Fragments of Genes IGF-1, FGF5, and KAP 1.4

The target fragments of genes *IGF-1*, *FGF5*, and *KAP 1.4* were amplified and cloned from Tibetan cashmere goats using gene-specific primers (Figure 2). After sequencing, we discovered that the cloned fragments of *IGF-1*, *FGF5*, and *KAP 1.4* were 320 bp, 193 bp, and 303 bp in length, respectively.

### 3.2. Sequence Alignment and the Phylogenetic Tree

The sequence alignments in the Tibetan cashmere goat, as well as other goat and sheep breeds, are shown in (Figure 3). The identicality of sequences in genes *FGF5* and *KAP 1.4* was 99% and in gene *IGF-1* was 98%. There were no significant differences among them. The sequences of genes *FGF5* and *KAP 1.4* were highly similar among Tibetan cashmere goats, other goat and sheep breeds. The sequences of genes *IGF-1*, *FGF5*, and *KAP 1.4* from other species were retrieved from GenBank, including *Capra hircus*, *Ovis aries*, *Bos taurus*, *Homo sapiens*, and *Mus musculus*.

### 3.3. Single Nucleotide Polymorphism Analysis

After amplification and sequence alignment of genes *IGF-1*, *FGF5*, and *KAP 1.4* in 206 Tibetan cashmere goats, two new SNP loci were discovered in gene *KAP 1.4*, but there were no SNP loci in the amplified fragments of genes *IGF-1* and *FGF5*. Furthermore, the sequence maps revealed that the two newly discovered SNPs were localized at 209 bp and 245 bp in the amplified fragment of *KAP 1.4*. The sequencing of 209 bp and 245 bp fragments revealed variable overlapping peak patterns (Figure 4). The genotype was split according to the nucleotides at 245 bp and 209 bp loci. There were three alleles at the 245 bp locus: C1C1, C1T1, and C2T2 (Figure 4A). However, only two genotypes were identified at the 209 bp locus: C2C2 and C2T2, because T2T2 was not detected (Figure 4B). When the sequences in each genotype were analyzed, both SNP loci were C–T mutations: CTG–TTG at 209 bp and CCT–CTT at 245 bp (Table 2). Moreover, the nucleotide mutation at 209 bp did not change the amino acid (Leucine–non-change), but the nucleotide mutation at 245 bp resulted in the transformation of the amino acid (Proline–Leucine).

Genotype frequencies and allele frequencies among the six counties are shown in Table 3 and Table 4, respectively. At the 245 bp locus, the average frequencies of genotypes C_1_C_1_, C_1_T_1_, and T_1_T_1_ were 0.76, 0.22, and 0.02, respectively, and the allele frequencies of C_1_ and T_1_ were 0.87 and 0.13, respectively. At the 209 bp locus the average frequencies of genotypes C_2_C_2_ and C_2_T_2_ were 0.94 and 0.06, respectively, and the allele frequencies of C_2_ and T_2_ were 0.97 and 0.03, respectively. The allele frequencies of T_1_ and T_2_ (C–T mutation) were the highest in Coqen county (0.245 and 0.055). In addition, all of the heterozygosity and homozygosity patterns showed differences in alleles and these alleles were specific to a genetic group.

### 3.4. Relative mRNA Expression Level of IGF-1, FGF5, and KAP 1.4

The mRNA expression levels of *IGF-1*, *FGF5*, and *KAP 1.4* were compared in different counties, sexes and ages of goats. As demonstrated in Figure 5A, the expression levels of the gene *IGF-1* in Baingoin and Nyima counties were significantly higher than in other counties (*p* < 0.05). Moreover, the expression level of gene *FGF5* in Gêrzê was significantly higher than in Rutog (Figure 5B). Comparison of mRNA expression level in *KAP 1.4* showed significant differences among the seven counties (Figure 5). Interestingly, there were no significant differences in mRNA levels between sexes in Tibetan cashmere goat genes *IGF-1*, *FGF5*, and *KAP 1.4* (Figure 6). Furthermore, mRNA expression differences in Tibetan cashmere goats were analysed by age. As shown in Figure 7, the mRNA expression levels of *IGF-1*, *FGF5*, and *KAP 1.4* were similar across one to five-year-old cashmere goats (*p* > 0.05), however the expression level of the gene *IGF-1* remained slightly up-regulated across one to five-year-old cashmere goats, except in 4-year-old cashmere goats. *FGF5* mRNA expression was lower in one and two-year-old goats compared with three to five-year-old goats. Furthermore, the expression levels of the gene *KAP 1.4* were progressively up-regulated from one to five-year-old cashmere goats.

## 4. Discussion

In China, the Tibetan cashmere goat is a valuable species, and its cashmere is extensively used in clothing and other textiles. *IGF-1*, *FGF5*, and *KAP 1.4* have been identified as key regulators in cashmere development. In this study, we investigated mRNA expression levels of these three genes in Tibetan cashmere goats and discovered *IGF-1*, *FGF5*, and *KAP 1.4* SNP loci. *IGF-1* has been identified as an essential growth hormone regulator that influences growth characteristics and yearling fleece in some goat breeds [5]. Previous research identified SNP loci of the *IGF-1* gene, which may serve as a molecular marker for growth traits in the Nanjiang Huang Goat [17]. Furthermore, it has been suggested that FGF5 may function as a negative regulator in hair development, inhibiting hair growth [18,27]. Previous research on *Ovis aries* found SNP loci in the *FGF5* gene, and the genotype and allele frequencies were significantly different between sheep breeds [11]. Other studies on the relationship between fiber growth and polymorphisms in the KAP family revealed that SNPs in the KAP gene definitely affected fiber quality and quantity [28].

Variation in *KRTAP20-1* has been linked to cashmere fiber crimped fiber length and weight [29,30]. A previous study compared size variability in the *KAP1.4* gene between *Ovis aries* and *Capra hircus* and discovered that the gene *KAP1.4* in sheep has a 30-nucleotide deletion. Furthermore, researchers discovered SNP loci in *KAP 1.4* fragments in Indian native goat breeds, but the associations of guard and down fiber length and diameter with various SNPs were non-significant (*p* > 0.05) [8]. Another study discovered that the SNP sites in *KAP 9.2* as well as the mRNA expression level of *KAP 9.2* were significantly different among genotypes of Shanbei white cashmere goats, Inner Mongolia white cashmere goats, and Guanzhong dairy goats. It also showed that the identified SNP may regulate translation or stability of *KAP 9.2* mRNA, which would be beneficial for marker-assisted selection in cashmere goat breeding [12]. A recent study found that SNP loci in genes *KAP 6* and *KAP 8* were associated with wool fiber diameter in Medium Peppin Merino sheep [24].

In accordance with previous research, particular exonic fragments of the genes *IGF-1*, *FGF5*, and *KAP 1.4* have been amplified, and SNPs in these genes may be associated with growth traits. As a result, we examined the relationship between mRNA expression levels and polymorphisms in the genes *IGF-1*, *FGF5*, and *KAP 1.4* and cashmere growth in Tibetan cashmere goats. Average annual cashmere production in different counties and ages of Tibetan cashmere goats is presented in Table 5 and Table 6. These tables show that the average cashmere production in Rutog, Coqen, and Nyima counties (350–380 g/each) is greater than that in the other three counties (300 g/each), and that cashmere production in one-year-old cashmere goats (50 g/each) is lower than that of two to five-year-old (50–200/each) cashmere goats. Two novel SNP loci in the *KAP 1.4* gene may affect cashmere production, and the mRNA expression levels of *KAP 1.4* and *FGF5* are also known to be associated with cashmere production. SNP detection results identified two SNP sites in *KAP 1.4*, but no SNPs were identified in *IGF-1* and *FGF5*. Among these two newly identified SNPs, only the SNP locus at 245 bp resulted in an amino acid change (Proline–Leucine).

According to Table 3, Table 4 and Table 5, both genotypes in SNP loci 245 bp and 209 bp revealed that Coqen county has the highest mutation rate. The average mutation rate at the SNP locus 245 bp is 24.3%, with the highest rate (42.9%) in Coqen county (350 g/each). As a result, it is hypothesized that this polymorphism in *KAP 1.4* is associated with cashmere growth, and that this mutation may contribute to a rise in Tibetan cashmere goat cashmere output. No SNPs were identified in the amplified fragment, implying that exons 1 and 4 of the *FGF5* gene did not appear in polymorphisms. Previous research has shown that SNPs in specific fragments of the genes *IGF-1* and *FGF5* occur [6,12], but in this study they were not detected in Tibetan cashmere goats. The difference between breeds may lead to this situation. The measured distance of various species in the phylogenetic tree indicated that the Tibetan cashmere goat was close to *Capra hircus*, *Bos taurus*, and *Ovis aries*, while *Homo sapiens* and *Mus musculus* were distinct species. The mRNA expression levels of genes *FGF5* and *KAP 1.4* were significantly different among the various counties (*p* < 0.05). The mRNA expression level of *IGF-1* is similar in most counties (*p* > 0.05) and similar to previous findings in other goat breeds [13]. The reasons for the relatively high expression levels of the gene *IGF-1* in Baingoin and Nyima could be due to individual variations in the animals or distinct living conditions. The counties of Baingoin and Nyima are parts of Nagqu Prefecture, while the remaining five counties are part of Ngari Prefecture. The average annual production of cashmere from Northwestern Tibetan cashmere goats in Rutog, Coqen, Nyima, Baingoin, Gê’gya, Gêrzê, and Gareach districts were 380, 350, 350, 300, 300, 300, and 300 g, respectively. The county with the highest cashmere output, Rutog (380 g/each), has the lowest *FGF5* mRNA expression level. This could imply that *FGF5* is a negative regulator in the hair cycle, inhibiting fiber development [18]. Counties with higher production of cashmere, Coqen and Nyima (350 g/each), had significantly higher mRNA expression levels of the gene *KAP* than other counties, including Rutog, Baingoin, Gê’gya, Gêrzê, and Gar. However, the expression levels in the top cashmere-producing area Rutog (380 g/each) are similar (*p* > 0.05). The habitat of the Tibetan cashmere goat, including altitude, climate, rearing conditions, and biological variety, may contribute to the low mRNA expression level of *KAP 1.4* in Rutog. The results of the mRNA expression levels of *IGF-1*, *FGF5,* and *KAP 1.4* by sex and year stages of Tibetan cashmere goats were similar (*p* > 0.05). However, there was a trend in Tibetan cashmere goat mRNA expression levels of *IGF-1* and *KAP 1.4*; these were up-regulated with age, with the exception of *IGF-1* in four-year-old cashmere goats. As shown in Table 6, cashmere production increased with age. *KAP 1.4* mRNA expression levels increase with age indicating that these may play a role in regulating cashmere development.

In this study, we analyzed whether the mRNA expression and polymorphisms of genes *IGF-1*, *FGF5*, and *KAP 1.4* were associated with cashmere production in Tibetan cashmere goats and we explored the possible features of these genes. The study of the relationship between cashmere production and the characteristics of these three genes revealed that *FGF5* and *KAP 1.4* may regulate cashmere production in Tibetan cashmere goats. Our results demonstrate that candidate genes of cashmere development in Tibetan cashmere goats may be used for analyzing cashmere quantity and quality. The detection of SNPs and the analysis of mRNA expression of genes *IGF-1*, *FGF5*, and *KAP 1.4* in the current research could provide data for genetic evolutionary analysis. Further research should be conducted into SNP detection of other fragments in genes *IGF-1* and *FGF5*, as well as signal pathways and gene functions in protein levels of genes *IGF-1*, *FGF5*, and *KAP 1.4* in Tibetan cashmere goats.

## Figures and Tables

**Figure 1 genes-14-00711-f001:**
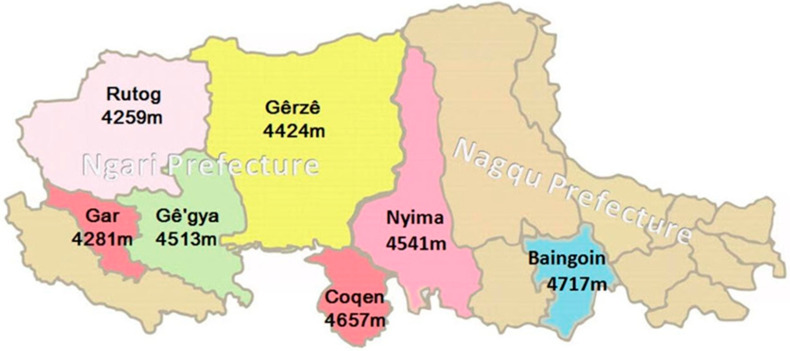
Distribution of the Northwestern Tibetan cashmere goat population, showing different counties and their altitudes.

**Figure 2 genes-14-00711-f002:**
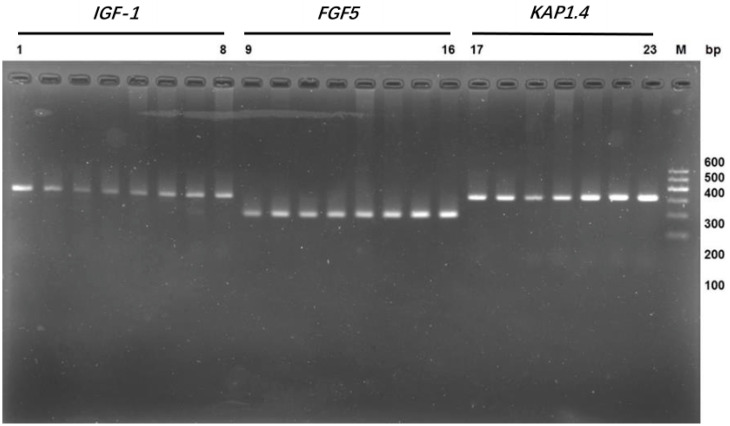
Amplification of genes *IGF-1*, *FGF5*, and *KAP 1.4* in Northwestern cashmere goats. Lanes 1–8: gene *IGF-1*; Lanes 9–16: gene *FGF5*; Lanes 17–23: gene *KAP 1.4*.

**Figure 3 genes-14-00711-f003:**
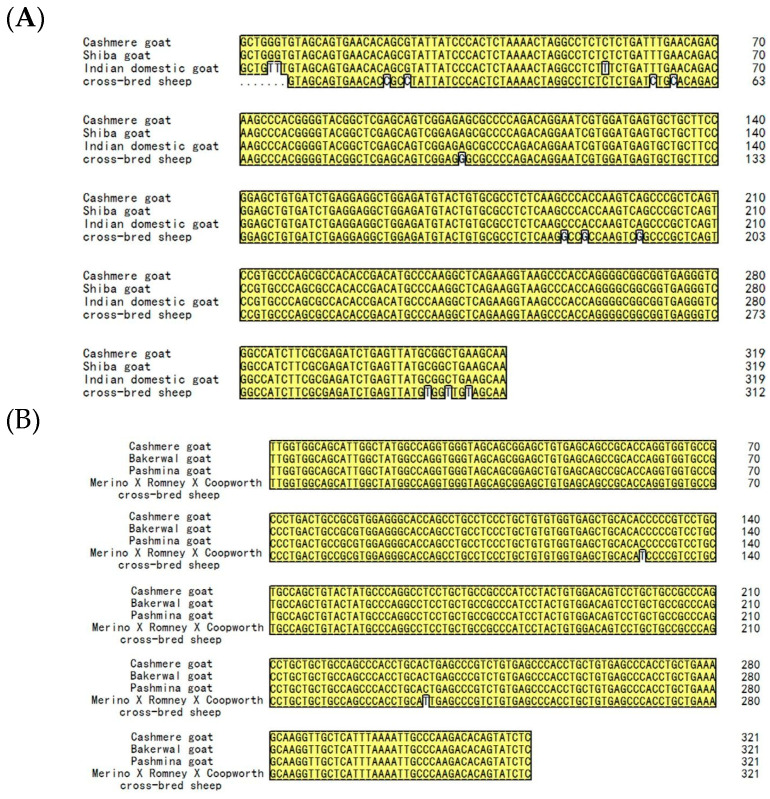
Nucleotide sequence alignment of genes *IGF-1* (**A**), *FGF5* (**B**), and *KAP 1.4* (**C**) in Northwestern Tibetan cashmere goats along with that of other goat and sheep breeds. “Cashmere goat” stands for Northwestern Tibetan cashmere goat.

**Figure 4 genes-14-00711-f004:**
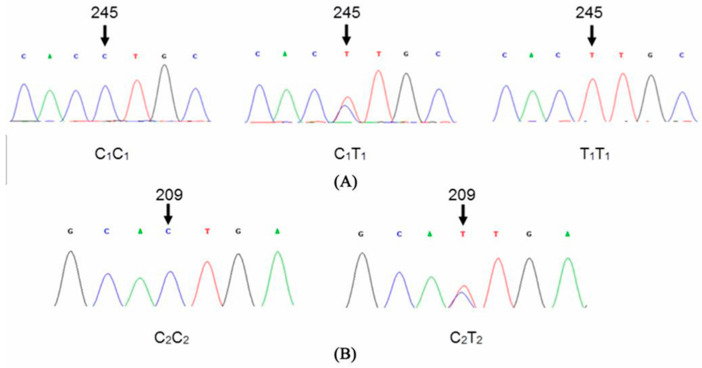
(**A**) Sequence maps of genotypes C_1_C_1_, C_1_T_1_, and T_1_T_1_ with g.245 C > T of gene *KAP1.4*. (**B**) Sequence maps of genotypes C_2_C_2_ and C_2_T_2_ with g.209 C > T of gene *KAP 1.4*.

**Figure 5 genes-14-00711-f005:**
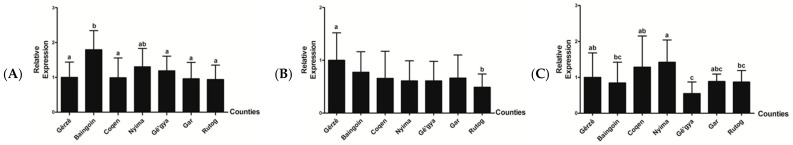
Relative mRNA expression of genes *IGF-1* (**A**), *FGF5* (**B**), and *KAP 1.4* (**C**) in ear skin tissue of the Northwestern Tibetan cashmere goat among seven counties. Data are mean ± SD from three independent experiments. Different letters stand for significant difference (*p* < 0.05).

**Figure 6 genes-14-00711-f006:**
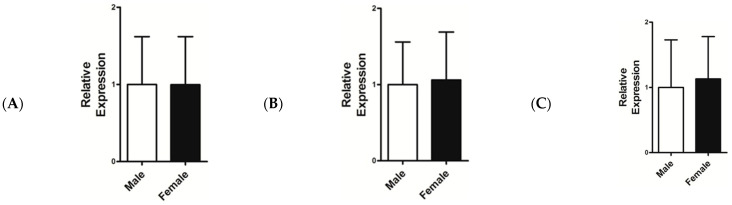
Relative mRNA expression of genes *IGF-1* (**A**), *FGF5* (**B**), and *KAP 1.4* (**C**) in ear skin tissue of the Northwestern Tibetan cashmere goat by sex. Data are mean ± SD from three independent experiments (*p* > 0.05).

**Figure 7 genes-14-00711-f007:**
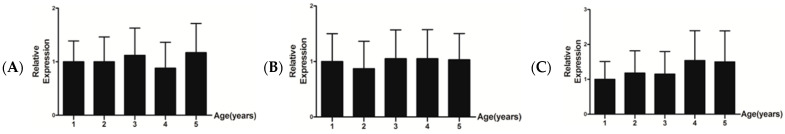
Relative mRNA expression of gene *IGF-1* (**A**), *FGF5* (**B**), and *KAP 1.4* (**C**) in ear skin tissue of the Northwestern Tibetan cashmere goat by age stages (one to five years old). Data are mean ± SD from three independent experiments (*p* > 0.05).

**Table 1 genes-14-00711-t001:** Primer sequences for PCR and real-time PCR.

Primer Name	Primer Sequence (5′-3′)	Product Size (bp)	Annealing Temperature (°C)	Accession Number
Primers for SNP detection				
IGF1-F	GCTGGGTGTAGCAGTGAACA	320	55.6	D26119
IGF1-R	GTTGCTTCAGCCGCATAACT			
FGF5-F	AGCAGTAGCACCGTGTCTTC	193	61.0	XM_01294679
FGF5-R	AGCCATTGACTTTGCCATCC			
KAP1.4-F	TTGGTGGCAGCATTGGCTATG	303	54.7	GQ507748
KAP1.4-R	AGAGATACTGTGTCTTGGGCA			
Primers for real time PCR				
IGF1-F	AATCAGCAGTCTTCCAACCCAA	114	51.8	NM_001285697
IGF1-R	AGCAAGCACAGGGCCAGAT			
FGF5-F	ACCTCAGCACGTCTCTACCCAC	123	52.4	KM_596772
FGF5-R	GGAACCTTTGGCTTGACGG			
KAP1.4-F	GCCAGCCAACTTCCATCCA	110	56.7	JQ627657
KAP1.4-R	AATGCCACAGCCGGTCTCAC			
β-action-F	GGCCGCACCACTGGCATTGTCAT	104	60.0	DQ845171.1
β-action-R	AGGTCCAGACGCAGGATGGCG			

*IGF-1*, Insulin-like Growth Factor I; *FGF5*, Fibroblast Growth Factor 5; *KAP1.4*, Keratin-associated protein 1.4; *β-actin*, beta actin.

**Table 2 genes-14-00711-t002:** Positions of mutations with changes in bases and their effects on amino acids in gene *KAP 1.4* of the Northwestern Tibetan cashmere goat.

Gene	Position of Mutation	Base Change	Codon	Amino Acid
*KAP1.4*	209	C–T	C*TG–TTG	Leucine(L)–non-change
	245	C–T	CC*T–CTT	Proline(P)–Leucine(L)

Locus of changed nucleotide.

**Table 3 genes-14-00711-t003:** Genotype and allele frequencies of gene *KAP 1.4* of the cashmere goat at the 245 bp locus.

Counties	Genotypes	Frequencies	Alleles	Frequencies	Heterozygosity	Homozygosity
C_1_C_1_		0.86	C_1_	0.930		
Gar (N = 37)	C1T1	0.14	T_1_	0.070	0.135	0.865
T_1_T_1_		0				
C_1_C_1_		0.84	C_1_	0.895		
Rutog (N = 37)	C1T1	0.11	T_1_	0.105	0.108	0.892
T_1_T_1_		0.05				
C_1_C_1_		0.73	C_1_	0.865		
Gê’gya (N = 30)	C1T1	0.27	T_1_	0.135	0.266	0.734
T_1_T_1_		0				
C_1_C_1_		0.78	C_1_	0.890		
Baingoin (N = 37)	C1T1	0.22	T_1_	0.110	0.216	0.784
T_1_T_1_		0				
C_1_C_1_		0.73	C_1_	0.845		
Gêrzê (N = 30)	C1T1	0.23	T_1_	0.155	0.233	0.767
T_1_T_1_		0.04				
C_1_C_1_		0.57	C_1_	0.755		
Coqen (N = 35)	C1T1	0.37	T_1_	0.245	0.371	0.629
T_1_T_1_		0.06				
C_1_C_1_ (N = 156)		0.76	C_1_	0.870		
Total (N = 206) Nyima C1T1 (N = 45)		0.22	T_1_	0.130	0.218	0.782
T_1_T_1_ (N = 5)		0.02				

**Table 4 genes-14-00711-t004:** Genotype and allele frequencies of gene *KAP 1.4* of the cashmere goat at the 209 bp locus. “N” represents the total number of cashmere goats in the counties.

Genetic Group	Genotypes	Frequencies	Alleles	Frequencies	Heterozygosity	Homozygosity
C_2_C_2_		0.97	C_2_	0.985		
Gar (N = 37)	C2T2	0.03	T_2_	0.015	0.027	0.973
T_2_T_2_		0				
C_2_C_2_		0.92	C_2_	0.960		
Rutog (N = 37)	C2T2	0.08	T_2_	0.040	0.081	0.919
T_2_T_2_		0				
C_2_C_2_		0.93	C_2_	0.965		
Gê’gya (N = 30)	C2T2	0.07	T_2_	0.035	0.066	0.934
T_2_T_2_		0				
C_2_C_2_		1	C_2_	1.000		
Baingoin (N = 37)	C2T2	0	T_2_	0.000	0.000	1.000
T_2_T_2_		0				
C_2_C_2_		0.93	C_2_	0.965		
Gêrzê (N = 30)	C2T2	0.07	T_2_	0.035	0.066	0.934
T_2_T_2_		0				
C_2_C_2_		0.89	C_2_	0.945		
Coqen (N = 35)	C2T2	0.11	T_2_	0.055	0.114	0.886
T_2_T_2_		0				
C_2_C_2_ (N = 194)		0.94	C_2_	0.970		
Total (N = 206)	C2T2 (N = 12)	0.06	T_2_	0.030	0.058	0.942
T_2_T_2_ (N = 0)		0				

**Table 5 genes-14-00711-t005:** Annual production of cashmere in each of the six districts with Northwestern Tibetan cashmere goats.

Counties	Rutog	Coqen	Nyima	Baingoin	Gê’gya	Gêrzê	Gar
Average cashmere production (g/each)	380	350	350	300	300	300	300

**Table 6 genes-14-00711-t006:** Annual production of cashmere in age 1 to 5-year-old Northwestern Tibetan cashmere goats.

Age	1-Year-Old	2-Year-Old	3-Year-Old	4-Year-Old	5-Year-Old
Average cashmere production (g/each)	<50	50–200	50–200	50–200	50–200

## Data Availability

Not applicable.

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
