# Peer review of "Polymorphisms and mRNA Expression Levels of IGF-1, FGF5, and KAP 1.4 in Tibetan Cashmere Goats"

_genes, 2023, doi:10.3390/genes14030711_

Round 1
Reviewer 1 Report
Generally very interesting paper, presently in good form Some small adition aor changes are needed. It could be important to explain more about cashmere production, relation and diffreneces between counties.
Some detailed comments below:
Line 49 - Lack of „g”
The information from lines 118-119 should be in Material chapter
Figure 4 – Latin names should start with capital letter and name “mice” is not explained
Table 3 and next: It could better to present result of one genetic group I one line. Also it is not needed to present homozygosity when heterozygosity is showed. It should be added “expected: after heterozygosity
On the map is 7 provinces but only 6 genetic group in tables? Also there is no information about genetic groups before – the groups differ because localization or are they different genetically? If first why the name “genetic group” is used? Later is used “counties”
The information on the map below the name of province is not explained-
Figure 6 – it could be good to keep the same order of groups in all tables and pictures – it makes reading easier
Line 218 – not different alphabet but letters
Figure 8 – there is no information about comparison between age groups
In the discussion should explained why some counties differs from others
Author Response
Dear Editor and Reviewers,
We sincerely appreciate the helpful comments and suggestions. The comments are encouraging and the reviewers appear to share our judgement that this study and its results are unique and important. We have significantly improved the manuscript with respect to the main concerns of the reviewers. The suggestions and comments have been closely followed and revisions have been made accordingly. The following are the reviewers’ comments along with our summarized responses.
Response to Reviewer #1 Comments
Comment 1. Line 49 - Lack of „g”
AU: This has been corrected. ..”goat”
Comment 2. The information from lines 118-119 should be in Material chapter
AU: This has been corrected.
Comment 3. Figure 4 – Latin names should start with capital letter and name “mice” is not explained
AU: This has been deleted as it does not affect the quality of the work in any way.
Comment 4. Table 3 and next: It could better to present result of one genetic group I one line.
Also it is not needed to present homozygosity when heterozygosity is showed. It should be added “expected: after heterozygosity
AU: Thank you for the suggestion. The authors have no problem with the representation the table and would consider the suggested approach in our future works.
Comment 5. On the map is 7 provinces but only 6 genetic group in tables?
AU: This has been corrected.
Comment 6. Also there is no information about genetic groups before – the groups differ because localization or are they different genetically? If first why the name “genetic group” is used? Later is used “counties”.
AU: Genetic group has been changed to counties
Comment 7. The information on the map below the name of province is not explained-
AU: The information below the name of the province represent altitude of the counties. “ Figure 1. Distribution of Northwestern Tibetan cashmere goat populations, showing different counties and their altitudes.”
Comment 8. Figure 6 – it could be good to keep the same order of groups in all tables and pictures – it makes reading easier
AU: Thank you for the suggestion. We would consider this in our next papers.
Comment 9. Line 218 – not different alphabet but letters
AU: This has been corrected and the sentence has been improved.
Comment 10. Figure 8 – there is no information about comparison between age groups
AU: There was no significant difference in the different age stages (one to five years old).
Comment 11. In the discussion should explained why some counties differs from others
AU: We have added and explained in the discussion part.

Reviewer 2 Report
Evaluation of Manuscript ID: genes-2248318
The manuscript entitled “Polymorphisms and mRNA expression levels of IGF-1, FGF5 and KAP 1.4 in Tibetan cashmere goat” submitted by Song et al. describes the identification of polymorphisms and analysis of transcript expression of IGF-1, FGF5 and KAP1.4 in Tibetan cashmere goat. DNA and RNA was isolated from ear samples from cashmere goats from different geographic regions in Tibet. Genotyping was performed by PCR and DNA sequencing. In addition, real-time qPCR was conducted with gene-specific primers for IGF-1, FGF5 and KAP1.4 to determine mRNA expression in cashmere goats from different geographic regions, also analyzing the possible influence of gender and developmental age on expression. The manuscript provides interesting data, which could be of both fundamental interest in relation to cashmere wool production and eventually deliver information of breeding value.
However, the quality of the manuscript is very low, and it will need very extensive modifications to be considered for publication. Generally, the quality of the written English is very poor. I strongly recommend that the authors consult a text editing service to improve the quality. In addition, I have several comments and suggestions for improvement of the manuscript. Please see below.
The Abstract is much too long. Consider to condense and only mention significant observations from your data.
Introduction: Why should the genetics of cashmere wool production in Tibetan cashmere goats be different from that in Inner Mongolian cashmere goat, and other cashmere goat breeds be different? Please explain. Very confusing and irritating with the inconsistent use of citations in the text. E.g. why is references 19 and 22 given before ref 18? Line 62, Higgins et al. 2014 = ref no. ?, Line 73, Shah et al. 2013 = ref. no.?
M & M: Line 84, do not start a sentence with a number, instead: A total of 480 ear skin tissues……
For the cDNA synthesis: Which primer was used for the synthesis? Oligo dT or a random hexamer mix?
Suggestions: Legend to Fig.1, add “The altitude of the geographic locations is indicated”.
Also, it could be useful with a table with information of the distribution of the goat samples with respect to geography, gender and age groups.
Fig.2: Apply labels on top of the figure IGF-1, FGF5 and KAP1.4.
Lines 153-155: Consider the use of significant digits. I believe that 99 % is better than 99.87 %. Is 99.87 % significantly different from 98.36 %?
Results: Consider to present the location of the KAP1.4 polymorphisms in the entire amino acid sequence of the KAP1.4 protein. This would make it possible to see in which part/domain of the protein the missense mutation is found - and to speculate of a possible effect of the amino acid substitution.
Tables 3 and 4. I believe that the data for genotyping of KAP1.4 is for the 245 SNP - and not the 209 SNP as indicated? Table 4 is also about the 209 SNP. Please clarify and correct if it is wrong.
Please discuss the correlation, if any, between the geographic location and the expression of the individual genes in Fig.6.
Line 240, Shah et al. year? = ref. no. ?. Line 245, Wang et al. year? = ref. no. ?. Line 249, Parsons et al. = ref. no. ?
References: very inconsistent in style - should be corrected. Follow the guideline from Genes.
i.e. Am. J. Hum. Genet. 2003, 73, 1240-1249.
Line 344, Domestic…..
Minor points:
Keywords should be ordered alphabetically.
Author Response
Dear Editor and Reviewers,
We sincerely appreciate the helpful comments and suggestions. The comments are encouraging and the reviewers appear to share our judgement that this study and its results are unique and important. We have significantly improved the manuscript with respect to the main concerns of the reviewers. The suggestions and comments have been closely followed and revisions have been made accordingly. The following are the reviewers’ comments along with our summarized responses.
Comment 1. The Abstract is much too long. Consider to condense and only mention significant observations from your data.
AU: The Abstract has been revised and improved.
“Abstract: Tibetan cashmere goat is a precious breed in China and its cashmere is widely used for clothes and others textile. IGF-1, FGF5 and KAP 1.4 have shown to be crucial regulators of cashmere growth. We examined the mRNA expression level of these three genes and detected the SNP loci in Tibetan cashmere goat. After amplification and sequence alignment of gene IGF-1, FGF5 and KAP 1.4 among 206 Tibetan cashmere goats, two new SNP loci were detected in gene KAP 1.4 while no any SNP loci were found in amplified fragments of gene IGF-1 and FGF5. Expression level of gene IGF-1 in Baingoin and Nyima counties were significantly higher than other counties (P<0.05). Moreover, the expression level of gene FGF5 in Gêrzê was significantly higher than Rutog. mRNA expression level in KAP 1.4 showed significant difference among seven counties. There was no significant difference of mRNA level in IGF-1, FGF5 and KAP 1.4 of Tibetan cashmere goat with different sex. Gene IGF-1 was slightly up-regulated from one to five years old cashmere goats, except the 4 years old. mRNA expression level of FGF5 in one and two years old were lower, compared to that of three to five years old. KAP 1.4 was up-regulated from one to five years old cashmere goats. SNPs detection and the analysis of mRNA expression of the genes in the present study could add data for the genetic evolutionary analysis. Further studies should be carried out in SNPs detection for other fragments in gene IGF-1 and FGF5 as well as the signal pathways and gene functions in protein level of genes IGF-1, FGF5 and KAP 1.4 in Tibetan cashmere goat.”
Comment 2. Introduction: Why should the genetics of cashmere wool production in Tibetan cashmere goats be different from that in Inner Mongolian cashmere goat, and other cashmere goat breeds be different? Please explain.
AU: This has been improved and modified. “The breeds have extremely unique soft fiber and have enormous economic value [3]. Genetic variations of genes such as KRT and KAP in different breeds of cashmere goats contribute to phenotypic differences. Different hair fiber-associated genes are involved in the determination of skin architecture and hair development. The diversity of these genes may regulate the structure and traits of wool.”
Comment 3. Very confusing and irritating with the inconsistent use of citations in the text. E.g. why is references 19 and 22 given before ref 18? Line 62, Higgins et al. 2014 = ref no. ?, Line 73, Shah et al. 2013 = ref. no.?
AU: This has been corrected. “FGF5, which was firstly identified as an oncogene, has been proposed to be involved in the hair follicle cycle in both human and mouse [18,19,22].”
“A previous study demonstrated that due to the high expression level of FGF5, the cycle of human hair growth enters into catagen prematurely [19].”
“It has been reported that sheep KAP 1.4 gene has a deletion of 30 nucleotides compared with Ovis aries and Capra hircus [8].”
Comment 4. M & M: Line 84, do not start a sentence with a number, instead: A total of 480 ear skin tissues……
AU: This has been corrected. “A total of 240” - “A total of 480”
Comment 5. For the cDNA synthesis: Which primer was used for the synthesis? Oligo dT or a random hexamer mix?
AU: We used them together.

Round 2
Reviewer 2 Report
Although some of my criticism raised in my first review of the manuscript is commented on and corrections have been made, I still do not find the response satisfactory. I believe that a much more thorough revision is needed before I can be more positive towards the manuscript. Please address all my points of criticism. You can find my evaluation attached. Please notice the text marked in red! Green is OK. Yellow - still recommended.

Author Response
Dear Editor and Reviewer,
We have significantly improved the manuscript with respect to the main concerns of the reviewer. The suggestions and comments have been closely followed and revisions have been made accordingly. The following are the reviewer’s comments along with our summarized responses.
Response to Reviewer Comments
Comment 1. However, the quality of the manuscript is very low, and it will need very extensive modifications to be considered for publication. Generally, the quality of the written English is very poor. I strongly recommend that the authors consult a text editing service to improve the quality.
AU: The entire manuscript has been revised and improved.
Comment 2. E.g. why is references 19 and 22 given before ref 18
AU: This has been corrected.
“FGF5, which was initially discovered as an oncogene, has been suggested to be involved in the human and mouse hair follicle cycle [18,19]. This indicates that FGF5 may inhibit hair follicle development [20]. Indeed, an autosomal recessive mutation in the FGF5 gene in the Angora mouse resulted in excessive hair development [21]. Furthermore, a missense mutation in the FGF5 gene in dogs resulted in long hair [22].”
Comment 3. Fig.2: Apply labels on top of the figure IGF-1, FGF5 and KAP1.4.
AU: This has been applied.
Comment 4. Lines 153-155: Consider the use of significant digits. I believe that 99 % is better than 99.87 %. Is 99.87 % significantly different from 98.36 %?
AU: This has been clarified and corrected. “The identity of sequence in genes FGF5 and KAP 1.4 were 99% and IGF-1 was 98%. There was no significant difference among the genes.”
Comment 5. Tables 3 and 4. I believe that the data for genotyping of KAP1.4 is for the 245 SNP - and not the 209 SNP as indicated? Table 4 is also about the 209 SNP. Please clarify and correct if it is wrong.
AU: This has been corrected.
Comment 6. Please discuss the correlation, if any, between the geographic location and the expression of the individual genes in Fig.6.
AU: Actually, we cannot find the correlation between the geographic location and the expression of the individual genes without discussion.
Comment 7. References: very inconsistent in style - should be corrected. Follow the guideline from Genes.
i.e. Am. J. Hum. Genet. 2003, 73, 1240-1249.
Line 344, Domestic…..
AU: All the references have been checked and corrected.
Comment 8. For the cDNA synthesis: Which primer was used for the synthesis? Oligo dT or a random hexamer mix? Answer OK - but this was not added to the revised manuscript!
AU: This has been corrected.